# Pay Attention to Features, Transfer Learn Faster CNNs

**Kafeng Wang**[*†1]**, Xitong Gao**[2*]**, Yiren Zhao**[3]**, Xingjian Li**[4]**, Dejing Dou**[5]**, Cheng-Zhong Xu**[6]

[1,2] Shenzhen Institutes of Advanced Technology, Chinese Academy of Sciences.
[1] University of Chinese Academy of Sciences. [3] University of Cambridge.
[4,5] Big Data Lab, Baidu Research. [6] University of Macau.
[1] `kf.wang@siat.ac.cn`, [2] `xt.gao@siat.ac.cn`.

## Abstract

Deep convolutional neural networks are now widely deployed in vision applications, but a limited size of training data can restrict their task performance. Transfer learning offers the chance for CNNs to learn with limited data samples by transferring knowledge from models pretrained on large datasets. Blindly transferring all learned features from the source dataset, however, brings unnecessary computation to CNNs on the target task. In this paper, we propose attentive feature distillation and selection (AFDS), which not only adjusts the strength of transfer learning regularization but also dynamically determines the important features to transfer. By deploying AFDS on ResNet-101, we achieved a state-of-the-art computation reduction at the same accuracy budget, outperforming all existing transfer learning methods. With a $10\times$ MACs reduction budget, a ResNet-101 equipped with AFDS transfer learned from ImageNet to Stanford Dogs 120, can achieve an accuracy 11.07% higher than its best competitor.

## 1 Introduction

Despite recent successes of CNNs achieving state-of-the-art performance in vision applications (Tan & Le, 2019; Cai & Vasconcelos, 2018; Zhao et al., 2018; Ren et al., 2015), there are two major shortcomings limiting their deployments in real life. First, training CNNs from random initializations to achieve high task accuracy generally requires a large amount of data that is expensive to collect. Second, CNNs are typically compute-intensive and memory-demanding, hindering their adoption to power-limited scenarios.

To address the former challenge, *transfer learning* (Pan & Yang, 2009) is thus designed to transfer knowledge learned from the source task to a target dataset that has limited data samples. In practice, we often choose a source dataset such that the input domain of the source comprises the domain of the target. A common paradigm for transfer learning is to train a model on a large source dataset, and then fine-tune the pre-trained weights with regularization methods on the target dataset (Zagoruyko & Komodakis, 2017; Yim et al., 2017; Li et al., 2018; Li & Hoiem, 2018; Li et al., 2019). For example, one regularization method, $L^2$-$SP$ (Li et al., 2018), penalizes the $L^2$-distances of pretrained weights on the source dataset and the weights being trained on the target dataset. The pretrained source weights serves as a starting point when training on the target data. During fine-tuning on the target dataset, the regularization constrains the search space around this starting point, which in turn prevents overfitting the target dataset.

Intuitively, the responsibility of transfer learning is to preserve the source knowledge acquired by important neurons. The neurons thereby retain their abilities to extract features from the source domain, and contribute to the network's performance on the target dataset.

---

[*]Equal contribution, corresponding authors.
[†]Work partially done during an internship at Baidu Research.

Moreover, by determining the importance of neurons, unimportant ones can further be removed from computation during inference with *network pruning* methods (Luo et al., 2017; He et al., 2017; Zhuang et al., 2018; Ye et al., 2018; Gao et al., 2019). The removal of unnecessary compute not only makes CNNs smaller in size but also reduces computational costs while minimizing possible accuracy degradations. As the source domain encompasses the target, many neurons responsible for extracting features from the source domain may become irrelevant to the target domain and can be removed. In Figure 1, a simple empirical study of the channel neurons' activation magnitudes corroborates our intuition: as deeper layers extract higher-level features, more neurons become either specialized or irrelevant to dogs. The discussion above hence prompts two questions regarding the neurons: *which neurons should we transfer source knowledge to, and which are actually important to the target model?*

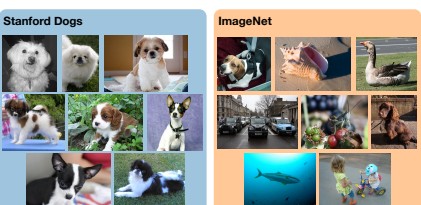 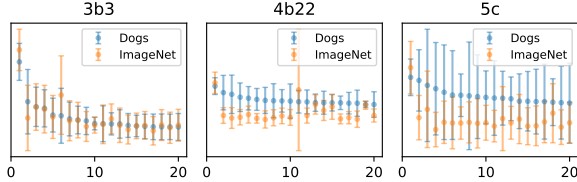

(a) Example images.          (b) Maximum channel activations.

Figure 1: (a) shows sample images from two datasets, ImageNet contains images with greater diversity. (b) shows the average maximum activations of 20 channel neurons in 3 layers of ResNet-101 that are most excited by images from Dogs.

Yet traditional transfer learning methods fail to provide answers to both, as generally they transfer knowledge either *equally* for each neuron with the same regularized weights, or determine the strength of regularization using only the source dataset (Li et al., 2018). The source domain could be vastly larger than the target, giving importance to weights that are irrelevant to the target task.

Recent years have seen a surge of interest in network pruning techniques, many of which induce sparsity by pushing neuron weights or outputs to zeros, allowing them to be pruned without a detrimental impact on the task accuracies. Even though pruning methods present a solution to neuron/weight importance, unfortunately they do not provide an answer to the latter question, *i.e.* whether these neurons/weights are important to the target dataset. The reason for this is that pruning optimization objectives are often in conflict with traditional transfer learning, as both drive weight values in different directions: zero for pruning and the initial starting point for transfer learning. As we will see later, a naïve composition of the two methods could have a disastrous impact on the accuracy of a pruned CNN transfer-learned on the target dataset.

In this paper, to tackle the challenge of jointly transferring source knowledge and pruning target CNNs, we propose a new method based on attention mechanism (Vaswani et al., 2017), *attentive feature distillation and selection* (AFDS). For the images in the target dataset, AFDS dynamically learns not only the features to transfer, but also the unimportant neurons to skip.

During transfer learning, instead of fine-tuning with $L^2$-SP regularization which explores the proximity of the pre-trained weights, we argue that a better alternative is to mimic the feature maps, *i.e.* the output response of each convolutional layer in the source model when images from the target dataset are shown, with $L^2$-distances. This way the fine-tuned model can still learn the behavior of the source model. Additionally, without the restriction of searching only the proximity of the initial position, the weights in the target model can be optimized freely and thus increasing their generalization capacity. Therefore, we present *attentive feature distillation* (AFD) to learn which relevant features to transfer.

To accelerate the transfer-learned model, we further propose *attentive feature selection* (AFS) to prune networks dynamically. AFS is designed to learn to predictively select important output channels in the convolution to evaluate and skip unimportant ones, depending

on the input to the convolution. Rarely activated channel neurons can further be removed from the network, reducing the model's memory footprint.

From an informal perspective, both AFD and AFS learn to adjust the "valves" that control the flow of information for each channel neuron. The former adjusts the strength of regularization, thereby tuning the flow of knowledge being transferred from the source model. The latter allows salient information to pass on to the subsequent layer and stops the flow of unimportant information. A significant attribute that differentiates AFD and AFS from their existing counterparts is that we employ attention mechanisms to adaptively learn to "turn the valves" dynamically with small trainable auxiliary networks.

Our main contributions are as follows:

- We present *attentive feature distillation and selection* (AFDS) to effectively transfer learn CNNs, and demonstrate state-of-the-art performance on many publicly available datasets with ResNet-101 (He et al., 2016) models transfer learned from ImageNet (Deng et al., 2009).

- We paired a large range of existing transfer learning and network pruning methods, and examined their abilities to trade-off FLOPs with task accuracy.

- By changing the fraction of channel neurons to skip for each convolution, AFDS can further accelerate the transfer learned models while minimizing the impact on task accuracy. We found that AFDS generally provides the best FLOPs and accuracy trade-off when compared to a broad range of paired methods.

## 2 RELATED WORK

### 2.1 TRANSFER LEARNING

Training a deep CNN to achieve high accuracy generally require a large amount of training data, which may be expensive to collect. *Transfer learning* (Pan & Yang, 2009) addresses this challenge by transferring knowledge learned on a large dataset that has a similar domain to the training dataset. A typical approach for CNNs is to first train the model on a large source dataset, and make use of their feature extraction abilities (Donahue et al., 2014; Razavian et al., 2014). Moreover, it has been demonstrated that the task accuracy can be further improved by fine-tuning the resulting pre-trained model on a smaller target dataset with a similar domain but a different task (Yosinski et al., 2014; Azizpour et al., 2015). Li et al. (2018) proposed $L^2$-SP regularization to minimize the $L^2$-distance between each fine-tuned parameter and its initial pre-trained value, thus preserving knowledge learned in the pre-trained model. In addition, they presented $L^2$-SP-Fisher, which further weighs each $L^2$-distance using Fisher information matrix estimated from the source dataset. Instead of constraining the parameter search space, Li et al. (2019) showed that it is often more effective to regularize feature maps during fine-tuning, and further learns which features to pay attention to. Learning without Forgetting (Li & Hoiem, 2018) learns to adapt the model to new tasks, while trying to match the output response on the original task of the original model using *knowledge distillation* (KD) (Hinton et al., 2014). Methods proposed by Zagoruyko & Komodakis (2017) and Yim et al. (2017) transfer knowledge from a teacher model to a student by regularizing features. The former computes and regularizes spatial statistics across all feature maps channels, whereas the latter estimates the flow of information across layers for each pair of channels, and transfers this knowledge to the student. Instead of manually deciding the regularization penalties and what to regularize as in the previous approaches, Jang et al. (2019) used meta-learning to automatically learn what knowledge to transfer from the teacher and to where in the student model.

Inspired by Li et al. (2019) and Jang et al. (2019), this paper introduces *attentive feature distillation* (AFD), which similarly transfers knowledge by learning from the teacher's feature maps. It however differs from Jang et al. (2019) as the teacher and student models share the same network topology, and it instead learns which channel to transfer from the teacher to the student in the same convolutional output.

## 2.2 Structured Sparsity

Sparsity in neural networks has been a long-studied subject (Reed, 1993; LeCun et al., 1990; Chauvin, 1989; Mozer & Smolensky, 1989; Hassibi et al., 1994). Related techniques have been applied to modern deep CNNs with great success (Guo et al., 2016; Dong et al., 2017a), significantly lowering their storage requirements. In general, as these methods zero out individual weights, producing irregular sparse connections, which cannot be efficiently exploited by GPUs to speed up computation.

For this, many recent work turned their attention to *structured sparsity* (Alvarez & Salzmann, 2016; Wen et al., 2016; Liu et al., 2017; He et al., 2017; 2018). This approach aims to find coarse-grained sparsity and preserves dense structures, thus allowing conventional GPUs to compute them efficiently. Alvarez & Salzmann (2016) and Wen et al. (2016) both added group Lasso to penalize non-zero weights, and removed channels entirely that have been reduced to zero. Liu et al. (2017) proposed *network slimming* (NS), which adds $L^1$ regularization to the trainable channel-wise scaling parameters $\gamma$ used in batch normalization, and gradually prunes channels with small $\gamma$ values by threshold. He et al. (2018) introduced *soft filter pruning* (SFP), which iteratively fine-tunes and sets channels with small $L^2$-norms to zero.

Pruning algorithms remove weights or neurons from the network. The network may therefore lose its ability to process some difficult inputs correctly, as the neurons responsible for them are permanently discarded. Gao et al. (2019) have found empirically that task accuracies degrades considerably when most of the computation are removed from the network, and introduced *feature boosting and suppression* (FBS). Instead of removing neurons permanently from the network, FBS learns to *dynamically* prune unimportant channels, depending on the current input image. In this paper, *attentive feature selection* (AFS) builds on top of the advantages of both static and dynamic pruning algorithms. AFS not only preserves neurons that are important to *some* input images, but also removes unimportant ones for *most* inputs from the network, reducing both the memory and compute requirements for inference.

There are methods that dynamically select which paths to evaluate in a network dependent on the input (Figurnov et al., 2017; Dong et al., 2017b; Bolukbasi et al., 2017; Lin et al., 2017; Shazeer et al., 2017; Wu et al., 2018; Ren et al., 2018). They however introduce architectural and/or training method changes, and thus cannot be applied directly on existing popular models pre-trained on ImageNet (Deng et al., 2009).

## 3 Attentive Feature Distillation and Selection

### 3.1 High-Level Overview

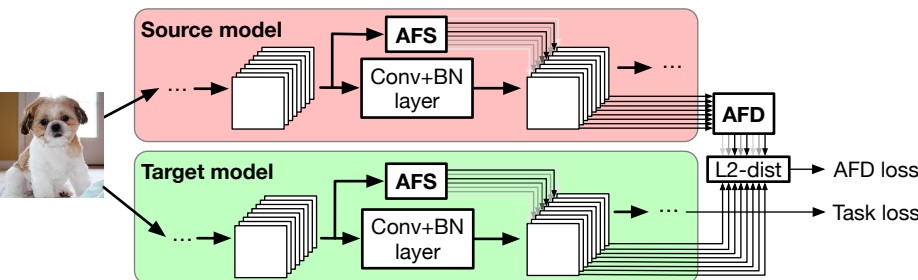

Figure 2: High-level overview of AFDS.

We begin by providing a high-level overview of attentive feature distillation and selection (AFDS). AFDS introduces two new components to augment each conventional *batch-normalized convolutional* (ConvBN) layer (Ioffe & Szegedy, 2015), as illustrated in Figure 2. The AFS preemptively learns the importance of each channel, in the output of the ConvBN layer, and can suppress unimportant channels, thus allowing the expensive convolution op-

eration to skip evaluating these channels. The AFD learns the importance of each channel in the output activation, and use the importance as weights to regularize feature maps in the target model with $L^2$-distance. Each component is a small neural network containing a small number of parameters that can be trained with conventional stochastic gradient descent (SGD).

## 3.2 Preliminaries

Consider a set of training data $\mathcal{D}$ where each sample $(\boldsymbol{x}, y)$ consists of an input image $\boldsymbol{x} \in \mathbb{R}^{C \times H \times W}$, and a ground-truth label $y \in \mathbb{N}$. Here $C$, $H$ and $W$ respectively denote the number of channels, and the height and width of the input image. Training a deep CNN classifier thus minimizes the following loss function with an optimization method based on SGD:

$$\mathcal{L}(\boldsymbol{\theta}) = \mathbb{E}_{(\boldsymbol{x},y)\sim\mathcal{D}}[\mathcal{L}^{\mathrm{CE}}(f(\boldsymbol{x},\boldsymbol{\theta}), y) + \mathcal{R}(\boldsymbol{\theta}, \boldsymbol{x}) + \lambda\|\boldsymbol{\theta}\|_2^2], \tag{1}$$

where $\boldsymbol{\theta}$ comprises all parameters of the model, the loss $\mathcal{L}^{\mathrm{CE}}(f(\boldsymbol{x},\boldsymbol{\theta}), y)$ denotes the cross-entropy loss between the CNN output $f(\boldsymbol{x},\boldsymbol{\theta})$ and the label $y$. The regularizer $\mathcal{R}(\boldsymbol{\theta}, \boldsymbol{x})$ is often used to reduce the risk of overfitting. In conventional training, $\mathcal{R}(\boldsymbol{\theta}, \boldsymbol{x}) = 0$. Finally, we impose a $L^2$ penalty on $\boldsymbol{\theta}$, where $\|\boldsymbol{z}\|_2$ represents the $L^2$-norm of $\boldsymbol{z}$ across all its elements.

We assume that $f(\boldsymbol{x},\boldsymbol{\theta})$ is a feed-forward CNN composed of $N$ ConvBN layers for feature extraction, $f_l(\boldsymbol{x}_{l-1}, \boldsymbol{\theta}_l)$ with $l \in L = \{1, 2, \ldots, N\}$, and a final fully-connected layer for classification, $g(\boldsymbol{x}_N, \boldsymbol{\theta}_g)$. Here, for the $l^{\mathrm{th}}$ layer, $\boldsymbol{x}_{l-1}$ is the input to the layer, with $\boldsymbol{x}_0$ indicating $\boldsymbol{x}$, and $\boldsymbol{\theta}_l$ is the layer's parameters. Therefore, the $l^{\mathrm{th}}$ layer is defined as:

$$\boldsymbol{x}_l = f_l(\boldsymbol{x}_{l-1}, \boldsymbol{\theta}_l) = \mathrm{relu}(\boldsymbol{\gamma}_l \cdot \mathrm{norm}(\mathrm{conv}(\boldsymbol{x}_{l-1}, \boldsymbol{\theta}_l)) + \boldsymbol{\beta}_l), \tag{2}$$

where $\boldsymbol{x}_l \in \mathbb{R}^{C_l \times H_l \times W_l}$ contains $C_l$ feature maps of the layer, each with a $H_l$ height and $W_l$ width. The function $\mathrm{conv}(\boldsymbol{x}_{l-1}, \boldsymbol{\theta}_l)$ is a convolution that takes $\boldsymbol{x}_{l-1}$ as input and uses trainable parameters $\boldsymbol{\theta}_l$, and $\mathrm{norm}(\boldsymbol{z})$ performs batch normalization. Finally, $\boldsymbol{\gamma}_l, \boldsymbol{\beta}_l \in \mathbb{R}^{C_l}$ are trainable vectors, the multiplications ($\cdot$) and additions ($+$) are channel-wise, and $\mathrm{relu}(\boldsymbol{z}) = \max(\boldsymbol{z}, 0)$ stands for the ReLU activation. Although we use the feed-forward classifier above for simplicity, it can be easily modified to contain additional structures such as residual connections (He et al., 2016) and computations for object detection (Ren et al., 2015).

During transfer learning, as we fine-tune the network with a different task, the final layer $g(\boldsymbol{x}_N, \boldsymbol{\theta}_g)$ is generally replaced with a new randomly-initialized one $h(\boldsymbol{x}_N, \boldsymbol{\theta}_h)$. To prevent overfitting, additional terms are used during transfer learning, for instance, $L^2$-SP (Li et al., 2018) further constrains the parameters $\boldsymbol{\theta}_l$ to explore around their initial values $\boldsymbol{\theta}_l^\star$:

$$\mathcal{R}(\boldsymbol{\theta}, \boldsymbol{x}) = \lambda_{\mathrm{SP}} \sum_{l \in L} \|\boldsymbol{\theta}_l - \boldsymbol{\theta}_l^\star\|_2^2 + \lambda_{\mathrm{L2}}\|\boldsymbol{\theta}\|_2^2. \tag{3}$$

Instead of regularizing parameters, methods based on *knowledge distillation* (Hinton et al., 2014) encourages the model to mimic the behavior of the original while learning the target task. *Learning without Forgetting* (LwF) (Li & Hoiem, 2018) uses the following regularizer to mimic the response from the original classifiers:

$$\mathcal{R}(\boldsymbol{\theta}, \boldsymbol{x}) = \lambda_{\mathrm{LwF}} \mathcal{L}^{\mathrm{CE}}(g^\star(f_L(\boldsymbol{x}, \boldsymbol{\theta}_L), \boldsymbol{\theta}_g^\star)), \tag{4}$$

where $f_L(\boldsymbol{x}, \boldsymbol{\theta}_L)$ indicates the first $N$ layers, and $g^\star$ and $\boldsymbol{\theta}_g^\star$ respectively denote the original fully-connected (FC) layer and its associated parameters, and generally $\lambda_{\mathrm{LwF}} = 1$. Zagoruyko & Komodakis (2017), Yim et al. (2017) and Li et al. (2019) chose to regularize feature maps in some intermediate layers $L' \subseteq L$. We assume that $\boldsymbol{x}_l^\star$ is the $l^{\mathrm{th}}$ layer output of the original model with weights $\boldsymbol{\theta}^\star$ when the input $\boldsymbol{x}$ is shown to the model, and $r$ is a method-dependent function that constrains the relationship between $\boldsymbol{x}_l^\star$ and $\boldsymbol{x}_l$. The regularizer can then be defined as follows:

$$\mathcal{R}(\boldsymbol{\theta}, \boldsymbol{x}) = \lambda_{\mathrm{KD}} \sum_{l \in L'} r(\boldsymbol{x}_l^\star, \boldsymbol{x}_l). \tag{5}$$

### 3.3 ATTENTIVE FEATURE DISTILLATION

A simple way to extend Equation (5) is to constrain the $L^2$-norm-distance between $\boldsymbol{x}_l^\star$ and $\boldsymbol{x}_l$, and thus pushing the target model to learn the feature map responses of the source:

$$\mathcal{R}(\boldsymbol{\theta}, \boldsymbol{x}) = \lambda_{\text{FD}} \sum_{l \in L'} \|\boldsymbol{x}_l^\star - \boldsymbol{x}_l\|_2^2. \tag{6}$$

The above formulation, however, places equal weight to each channel neurons of the feature maps. As we discussed earlier, the importance of channel neurons varies drastically when different input images are shown. it is thus desirable to enforce a different penalty for each channel depending on the input $\boldsymbol{x}$. For this purpose, we design the regularizer:

$$\mathcal{R}(\boldsymbol{\theta}, \boldsymbol{x}) = \lambda_{\text{AFD}} \sum_{l \in L'} \sum_{c \in C_l} \boldsymbol{\rho}_l^{[c]}(\boldsymbol{x}_l^\star) \|(\boldsymbol{x}_l^\star - \boldsymbol{x}_l)^{[c]}\|_2^2. \tag{7}$$

Note that in Equation (7), for any tensor $\boldsymbol{z}$, the term $\boldsymbol{z}^{[c]}$ denotes the $c^{\text{th}}$ slice of the tensor. The transfer importance predictor $\boldsymbol{\rho}_l : \mathbb{R}^{C_l \times H_l \times W_l} \to \mathbb{R}^{C_l}$ computes for each channel the importance of the source activation maps, which governs the strength of the $L^2$ regularization for each channel. The predictor function is trainable and is defined as a small network with two FC layers:

$$\boldsymbol{\rho}_l^{[c]}(\boldsymbol{x}_l^\star) = \text{softmax}(\text{relu}(\flat(\boldsymbol{x}_l^\star)\boldsymbol{\varphi}_l + \boldsymbol{\nu}_l)\,\boldsymbol{\varphi}_l' + \boldsymbol{\nu}_l'). \tag{8}$$

The function $\flat : \mathbb{R}^{C \times H \times W} \to \mathbb{R}^{C \times HW}$ flattens the spatial dimensions in a channel-wise fashion; The parameters $\boldsymbol{\varphi}_l \in \mathbb{R}^{HW \times H}$, $\boldsymbol{\nu}_l \in \mathbb{R}^{1 \times H}$, $\boldsymbol{\varphi}_l' \in \mathbb{R}^H$ and $\boldsymbol{\nu}_l' \in \mathbb{R}^C$ can thus be trained to adjust the importance of each channel dynamically; finally, the softmax activation is borrowed from attention mechanism (Vaswani et al., 2017) to normalize the importance values. In our experiments, $\boldsymbol{\varphi}_l$ and $\boldsymbol{\varphi}_l'$ use He et al. (2015)'s initialization, $\boldsymbol{\nu}_l$ and $\boldsymbol{\nu}_l'$ are both initialized to $\mathbf{0}$.

### 3.4 ATTENTIVE FEATURE SELECTION

In a fashion similar to *feature boosting and suppression* (FBS) (Gao et al., 2019), AFS modifies the ConvBN layers from Equation (2):

$$\hat{f}_l(\boldsymbol{x}_{l-1}, \boldsymbol{\theta}_l) = \text{relu}(\boldsymbol{\pi}_l(\boldsymbol{x}_{l-1}) \cdot \text{norm}(\text{conv}(\boldsymbol{x}_{l-1}, \boldsymbol{\theta}_l)) + \boldsymbol{\beta}_l), \tag{9}$$

where the predictor function takes as input the activation maps of the previous layer, *i.e.* $\boldsymbol{\pi}_l : \mathbb{R}^{C_{l-1} \times H_{l-1} \times W_{l-1}} \to \mathbb{R}^C$, is used to replace the vector $\boldsymbol{\gamma}_l$. This function dynamically predicts the importance of each channel, and suppresses certain unimportant channels by setting them to zero. The expensive conv function can hence be accelerated by skipping the disabled output channels. The predictor function is defined as below:

$$\boldsymbol{\pi}_l(\boldsymbol{x}_{l-1}) = \mathbf{m}_l \cdot q_l(\boldsymbol{x}_{l-1}), \text{ where } q_l(\boldsymbol{x}_{l-1}) = \text{wta}_{\lceil dC_l \rceil}(\mathbf{s}_l \cdot h_l(\boldsymbol{x}_{l-1}) + (1 - \mathbf{s}_l) \cdot \boldsymbol{\gamma}_l), \tag{10}$$

where $\mathbf{m}_l, \mathbf{s}_l \in \{0, 1\}^{C_l}$ are both constant masks that take binary values: $\mathbf{m}_l$ prunes output channels by permanently setting them to zeros, and $\mathbf{s}_l$ decides for each channel whether the output of $h_l(\boldsymbol{x}_{l-1})$ or $\boldsymbol{\gamma}_l$ should be used. It is clear that when $\mathbf{m}_l = \mathbf{1}$, no channel neurons are removed from the network. In Section 3.5, we explain how $\mathbf{m}_l$ and $\boldsymbol{\gamma}_l$ can be determined during the fine-tuning process. The *winner-take-all* function $\text{wta}_{\lceil dC_l \rceil}(\boldsymbol{z})$ preserves the $\lceil dC_l \rceil$ most salient values in $\boldsymbol{z}$, and suppresses the remaining ones by setting them to zeros. The density value $0 < d \leq 1$ is a constant that controls the number of channels to preserve during inference, with 1 preserving all $C_l$ channels. The smaller $d$ gets, the more channels can be skipped, which in turn accelerates the model. Finally, the function $h_l : \mathbb{R}^{C_{l-1} \times H \times W} \to \mathbb{R}^{C_l}$ is a small network that is used to predict the importance of each channel. It is composed of a global average pool followed by a FC layer, where $\text{pool} : \mathbb{R}^{C_{l-1} \times H \times W} \to \mathbb{R}^{C_{l-1}}$ computes the average across the spatial dimensions for each channel:

$$h(\boldsymbol{x}_{l-1}) = \text{relu}(\text{pool}(\boldsymbol{x}_{l-1})\boldsymbol{\varphi}_l'' + \boldsymbol{\nu}_l''). \tag{11}$$

For the initialization of the FC parameters, we apply He et al. (2015)'s method on the trainable weights $\boldsymbol{\varphi}_l'' \in \mathbb{R}^{C_{l-1} \times C_l}$ and $\boldsymbol{\nu}_l'' \in \mathbb{R}^{C_l}$ is initialized to zeros.

### 3.5 Training Procedure

In this section, we describe the pipeline of AFDS for transferring knowledge from a source model to a new model by fine-tuning on target dataset. The detailed algorithm can be found in Appendix A.

Initially, we have a pre-trained model $f$ with parameters $\theta^\star$ for the source dataset (*e.g.* ImageNet). To ensure better accuracies on compressed target models, All ConvBN layers $f_l$ in $f$ are extended with AFS as discussed in Section 3.4, with $d$ initially set to 1, which means that all output channels in a convolutional layer are evaluated during inference, *i.e.* no acceleration. The pre-trained model is then fine-tuned on the target training dataset $\mathcal{D}$ with the AFD regularization proposed in Section 3.3.

Empirically we found that in residual networks with greater depths, AFS could become notably challenging to train to high accuracies. To mitigate this, for each output channel of a layer $l$ we update $\mathbf{s}_l$ according to the variance of $h_l(\boldsymbol{x}_{l-1})$ observed on the target dataset. For each channel if the variance is smaller than a threshold $\delta_s$, then we set the entry in $\mathbf{s}_l$ to zero for that particular channel. This action replaces the output of $h_l(\boldsymbol{x}_{l-1})$ with $\boldsymbol{\gamma}_l$, which is a trainable parameter initialized to the mean of $h_l(\boldsymbol{x}_{l-1})$. We compute the mean and variance statistics using Welford (1962)'s online algorithm which can efficiently compute the statistics in a single-pass with $O(1)$ storage. In our experiments, $\delta_s$ is set to a value such that 50% of the channel neurons use the predictor function $h_l$.

Moreover, we discovered that many of the channel neurons are rarely activated in a AFS-based network. We further propose to remove the channel neurons that are activated with a low frequency. In each layer $l$, the mask $\mathbf{m}_l$ is used to disable certain channels from the network by setting their output to a constant $\mathbf{0}$, if the probability of a channel neuron being active is lower than $\delta_m$. Zeroed-out channels can thus be permanently removed when the model is used in inference.

## 4 Experiments

In this section we provide an extensive empirical study of the joint methods of transfer learning and channel pruning. We evaluate the methods with 6 different benchmark datasets: *Caltech-256* (Griffin et al., 2007) of 256 general object categories; *Stanford Dogs 120* (Khosla et al., 2011) specializes to images containing dogs; *MIT Indoors 67* (Quattoni & Torralba, 2009) for indoor scene classification; *Caltech-UCSD Birds-200-2011* (CUB-200-2011) (Wah et al., 2011) for classifying birds; and *Food-101* (Bossard et al., 2014) for food categories. We refer to Li et al. (2018) and Li et al. (2019), for a detailed description of the benchmark datasets. For Caltech-256, we randomly sample either 30 or 60 images from the training set for each category to produce *Caltech-256-30* and *-60* training datasets.

We use the ResNet-101 from torchvision[1] pre-trained on ImageNet as the network for experiments. For ResNet-101 equipped with AFS, we start by extending the pre-trained model and replacing each batch normalization with a randomly initialized AFS, and fine-tune the resulting model on ImageNet for 90 epochs with a learning rate of 0.01 decaying by a factor of 10 every 30 epochs. The resulting model matches its original baseline accuracy.

For each benchmark dataset, the final FC layer of the network is replaced with a new FC randomly initialized with He et al. (2015)'s method to match the number of output categories accordingly. We then perform transfer learning with 4 different methods: $L^2$ (fine-tuning without additional regularization), $L^2$-SP (Li et al., 2018), learning without forgetting (LwF) (Li & Hoiem, 2018), and finally AFD for models using AFS.

To accelerate the resulting fine-tuned models, we continue fine-tuning the model while gradually pruning away channels used during inference. For this, we separately examine 3 pruning strategies: network slimming (NS) (Liu et al., 2017), soft filter pruning (SFP) (He et al., 2018) and finally AFS for models transfer learned with AFD. Note that NS prunes channels by sorting them globally, while SFP does so in a layer-wise manner with identical prune

---

[1]https://pytorch.org/docs/stable/torchvision/index.html

ratios. During this procedure, we start with an unpruned model and incrementally remove 10% of the channels used in inference, *i.e.* preserving 90%, 80%, and *etc.*, down to 10% of all channels for the accelerated models. At each step, we fine-tune each model using 4500 steps of SGD with a batch size of 48, at a learning rate of 0.01, before fine-tuning for a further 4500 steps at a learning rate of 0.001. AFS additionally updates the **m** and **s** masks between the two fine-tuning runs.

Table 1: Top-1 accuracy (%) comparisons of NS, SFP and AFDS on 6 datasets fine-tuned with their respective best transfer learning methods under various speed-up constraints.

| MACs reduction | | NS | SFP | AFDS |
|---|---|---|---|---|
| MIT Indoors 67 | 2× | $81.83 \pm 0.35$ | $79.43 \pm 0.50$ | $\mathbf{82.05 \pm 0.43}$ |
| | 5× | $69.38 \pm 0.27$ | $60.43 \pm 0.31$ | $\mathbf{69.93 \pm 0.52}$ |
| | 10× | $1.50 \pm 0.30$ | $58.49 \pm 0.34$ | $\mathbf{66.72 \pm 0.53}$ |
| Stanford Dogs 120 | 2× | $87.21 \pm 0.58$ | $81.74 \pm 0.26$ | $\mathbf{87.41 \pm 0.56}$ |
| | 5× | $73.44 \pm 0.27$ | $61.20 \pm 0.31$ | $\mathbf{75.14 \pm 0.52}$ |
| | 10× | $1.33 \pm 0.50$ | $59.63 \pm 0.23$ | $\mathbf{70.70 \pm 0.33}$ |
| Caltech-256-30 | 2× | $\mathbf{85.87 \pm 0.38}$ | $77.26 \pm 0.28$ | $85.15 \pm 0.75$ |
| | 5× | $66.57 \pm 0.23$ | $64.27 \pm 0.31$ | $\mathbf{66.64 \pm 0.32}$ |
| | 10× | $0.39 \pm 0.04$ | $57.11 \pm 0.54$ | $\mathbf{61.45 \pm 0.43}$ |
| Caltech-256-60 | 2× | $\mathbf{88.02 \pm 0.45}$ | $84.59 \pm 0.28$ | $87.15 \pm 0.75$ |
| | 5× | $73.95 \pm 0.27$ | $68.38 \pm 0.59$ | $\mathbf{74.46 \pm 0.52}$ |
| | 10× | $5.05 \pm 0.11$ | $61.27 \pm 0.49$ | $\mathbf{70.16 \pm 0.53}$ |
| CUB-200-2011 | 2× | $\mathbf{78.88 \pm 0.65}$ | $75.65 \pm 0.26$ | $78.03 \pm 0.45$ |
| | 5× | $\mathbf{73.44 \pm 0.27}$ | $61.50 \pm 0.31$ | $73.35 \pm 0.52$ |
| | 10× | $0.52 \pm 0.50$ | $57.88 \pm 0.23$ | $\mathbf{69.07 \pm 0.43}$ |
| Food-101 | 2× | $83.78 \pm 0.61$ | $75.65 \pm 0.26$ | $\mathbf{84.21 \pm 0.65}$ |
| | 5× | $73.36 \pm 0.45$ | $17.10 \pm 0.17$ | $\mathbf{79.12 \pm 0.52}$ |
| | 10× | $0.99 \pm 0.04$ | $3.85 \pm 0.09$ | $\mathbf{76.95 \pm 0.49}$ |

Table 2: Top-1 accuracy (%) comparisons of $L^2$, $L^2$-SP, LwF, AFDS on 6 datasets fine-tuned with their respective best pruning methods under various speed-up constraints.

| MACs reduction | | $L^2$ | $L^2$-SP | LwF | AFDS |
|---|---|---|---|---|---|
| MIT Indoors 67 | 2× | $79.13 \pm 0.16$ | $78.09 \pm 0.56$ | $81.83 \pm 0.35$ | $\mathbf{82.05 \pm 0.43}$ |
| | 5× | $64.02 \pm 0.21$ | $62.00 \pm 0.31$ | $69.38 \pm 0.27$ | $\mathbf{69.93 \pm 0.52}$ |
| | 10× | $58.04 \pm 0.38$ | $58.49 \pm 0.34$ | $48.09 \pm 0.52$ | $\mathbf{66.72 \pm 0.53}$ |
| Stanford Dogs 120 | 2× | $85.38 \pm 0.67$ | $87.21 \pm 0.58$ | $87.07 \pm 0.35$ | $\mathbf{87.41 \pm 0.56}$ |
| | 5× | $70.20 \pm 0.37$ | $67.10 \pm 0.31$ | $73.44 \pm 0.27$ | $\mathbf{75.14 \pm 0.52}$ |
| | 10× | $59.63 \pm 0.23$ | $42.89 \pm 0.48$ | $17.79 \pm 0.50$ | $\mathbf{70.70 \pm 0.33}$ |
| Caltech-256-30 | 2× | $83.83 \pm 0.62$ | $83.67 \pm 0.53$ | $\mathbf{85.87 \pm 0.38}$ | $85.15 \pm 0.75$ |
| | 5× | $61.45 \pm 0.17$ | $60.03 \pm 0.21$ | $66.57 \pm 0.23$ | $\mathbf{66.64 \pm 0.32}$ |
| | 10× | $57.11 \pm 0.54$ | $56.12 \pm 0.31$ | $40.32 \pm 0.34$ | $\mathbf{61.45 \pm 0.43}$ |
| Caltech-256-60 | 2× | $86.27 \pm 0.47$ | $85.84 \pm 0.51$ | $\mathbf{88.02 \pm 0.45}$ | $87.15 \pm 0.75$ |
| | 5× | $71.02 \pm 0.37$ | $69.9 \pm 0.31$ | $73.95 \pm 0.27$ | $\mathbf{74.46 \pm 0.52}$ |
| | 10× | $61.27 \pm 0.49$ | $39.41 \pm 0.71$ | $26.75 \pm 0.50$ | $\mathbf{70.16 \pm 0.53}$ |
| CUB-200-2011 | 2× | $76.27 \pm 0.37$ | $75.58 \pm 0.46$ | $\mathbf{78.88 \pm 0.65}$ | $78.03 \pm 0.45$ |
| | 5× | $66.48 \pm 0.37$ | $64.49 \pm 0.31$ | $\mathbf{73.44 \pm 0.27}$ | $73.35 \pm 0.52$ |
| | 10× | $57.88 \pm 0.23$ | $57.13 \pm 0.38$ | $29.57 \pm 0.31$ | $\mathbf{69.07 \pm 0.43}$ |
| Food-101 | 2× | $83.78 \pm 0.61$ | $82.27 \pm 0.23$ | $82.38 \pm 0.85$ | $\mathbf{84.21 \pm 0.65}$ |
| | 5× | $73.36 \pm 0.33$ | $70.12 \pm 0.71$ | $73.05 \pm 0.64$ | $\mathbf{79.12 \pm 0.52}$ |
| | 10× | $1.6 \pm 0.04$ | $3.56 \pm 0.08$ | $3.85 \pm 0.09$ | $\mathbf{76.95 \pm 0.49}$ |

For each pruned model, we can compute the number of *multiply-accumulate operations* (MACs) required to perform inference on an image. For each accelerated convolution, the required number of MACs is $k^2 HWC_{\mathrm{in}}C_{\mathrm{out}}$, where $C_{\mathrm{in}}$ and $C_{\mathrm{out}}$ are the number of input and output channels that are not pruned, respectively. We compute the total number of MACs by summing up the MACs in all convolutions, residual connections, and the final pooling and FC layers. For AFS as we dynamically select which channels to evaluate during inference, we additionally add the overhead of the importance predictor layers to the number of total MACs.

Table 3: Comparison to related transfer learning methods.

| Dataset | Method | Model | Accuracy | MACs |
|---------|--------|-------|----------|------|
| CUB-200-2011 | Zagoruyko & Komodakis (2017) | ResNet-34 | 73.5 | 3.6 G |
| | | ResNet-18 | 73.0 | 1.8 G |
| | Jang et al. (2019) | ResNet-18 | 65.05 | 1.8 G |
| | AFDS | ResNet-101 | 76.34 | 2.4 G |
| | | ResNet-101 | 73.35 | 1.9 G |
| MIT Indoors 67 | Zagoruyko & Komodakis (2017) | ResNet-34 | 74.0 | 3.6 G |
| | | ResNet-18 | 72.9 | 1.8 G |
| | Jang et al. (2019) | ResNet-18 | 64.85 | 1.8 G |
| | AFDS | ResNet-101 | 78.09 | 2.4 G |
| | | ResNet-101 | 74.57 | 1.9 G |

(a) Stanford Dogs 120.     (b) Caltech-256-60.

Figure 3: MACs and accuracy (%) trade-off comparisons among different joint methods.

In Figure 3, we present the trade-off relationship between the number of *vs.* the target dataset accuracies for Stanford Dogs and Caltech-256-60. It is clear that AFDS (ours) exceeds various combinations of pruning methods (NS, SFP) and transfer learning methods ($L^2$, $L^2$-SP, LwF). The results for the remaining datasets can be found in Appendix B. The trade-off curves show that AFDS minimizes accuracy degradation even if 47% of the total MACs are removed from the original model, AFDS resulted in only 1.83% drop in accuracy for the model trained on Stanford Dogs. In extreme cases where we permit only $\frac{1}{10}$ of the original computations, our method can still manage a 70.70% accuracy, which is substantially better when compared to other pruning algorithms: NS drops to 1.33% and SFP only has 59.63%.

Table 1 provide numerical comparisons of different pruning methods against AFS under various speed-up constraints. Table 2 similarly compares transfer learning strategies against AFD. Under most acceleration requirements, the combined method, AFDS, achieves the best accuracies on the target datasets. Finally, Table 3 compares AFDS against other literatures that performs transfer learning. AFDS can achieve state-of-the-art accuracies when compared to methods that produce models with similar number of MACs.

## 5 CONCLUSION

In this paper, we introduced attentive feature distillation and selection (AFDS), a dual-attention method that aims to reap the advantages of transfer learning and channel pruning methods. By applying AFDS during fine-tuning, we can not only learn a new model with a higher target task accuracy, but also further accelerates it by computing a subset of channel neurons in each convolutional layers. Under a wide range of datasets, we demonstrated the smallest drop in validation accuracies under the same speed-up constraints when compared to traditional compression methods such as network slimming (Liu et al., 2017) and soft filter pruning (He et al., 2018).

ACKNOWLEDGEMENTS

This work is supported in part by National Key R&D Program of China (No. 2019YFB2102100), Science and Technology Development Fund of Macao S.A.R (FDCT) under number 0015/2019/AKP, Shenzhen Discipline Construction Project for Urban Computing and Data Intelligence, the National Natural Science Foundation of China (Nos. 61806192, 61802387), Shenzhen Science and Technology Innovation Commission (No. JCYJ2017081853518789, JCYJ20190812160003719), the Guangdong Science and Technology Plan Guangdong-Hong Kong Cooperation Innovation Platform (No. 2018B050502009), and China's Post-doctoral Science Fund (No. 2019M663183).

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

## A  THE OVERALL TRAINING ALGORITHM

In Algorithm 1 we illustrate the complete training procedure described above. Here, the function takes as input the target training dataset $\mathcal{D}$, the source model $f$ and its parameters $\boldsymbol{\theta}^\star$, the total number of steps to fine-tune $S$, the initial learning rate $\alpha$, and the threshold hyperparameters $\delta_\mathrm{s}$ and $\delta_\mathrm{m}$ respectively for $\mathbf{s}_l$ and $\mathbf{m}_l$. The function returns the optimized parameters $\boldsymbol{\theta}$ for the target dataset, and both constant masks for all layers $\mathbf{s} = (\mathbf{s}_1, \mathbf{s}_2, \ldots, \mathbf{s}_L)$ and $\mathbf{m} = (\mathbf{m}_1, \mathbf{m}_2, \ldots, \mathbf{m}_L)$. The function SGD then fine-tunes the model parameters. For each layer $l$, we compute the mean $\boldsymbol{\mu}_l$ and variance $\boldsymbol{\sigma}_l$ statistics of $q_l(\boldsymbol{x}_{l-1})$, and use it to compute $\mathbf{s}_l$.

---

**Algorithm 1** Training Procedure

---
1: **function** AFDS($\mathcal{D}, f, \boldsymbol{\theta}^\star, S, \alpha, \delta_\mathrm{s}, \delta_\mathrm{m}$)
2:    **for** $l \in L : \mathbf{s}_l \leftarrow \mathbf{1}$
3:    **for** $l \in L : \mathbf{m}_l \leftarrow \mathbf{1}$
4:    $\boldsymbol{\theta} \leftarrow \mathrm{SGD}(\mathcal{D}, f, \boldsymbol{\theta}^\star, \mathbf{s}, \mathbf{m}, \lceil \frac{S}{2} \rceil, \alpha, \mathcal{R})$
5:    **for** $l \in L$ **do**
6:        $\boldsymbol{\mu}_l \leftarrow \mathbb{E}_{(\boldsymbol{x},y)\sim\mathcal{D}}[q_l(\boldsymbol{x}_{l-1})]$
7:        $\boldsymbol{\sigma}_l^2 \leftarrow \mathbb{E}_{(\boldsymbol{x},y)\sim\mathcal{D}}[(q_l(\boldsymbol{x}_{l-1}) - \boldsymbol{\mu}_l)^2]$
8:        $\mathbf{p}_l \leftarrow \mathbb{E}_{(\boldsymbol{x},y)\sim\mathcal{D}}[\boldsymbol{\pi}_l(\boldsymbol{x}_{l-1}) > 0]$
9:        $\mathbf{s}_l \leftarrow \boldsymbol{\sigma}_l^2 > \delta_\mathrm{s}$
10:       $\boldsymbol{\gamma}_l \leftarrow \boldsymbol{\mu}_l$
11:       $\mathbf{m}_l \leftarrow \mathbf{p}_l > \delta_\mathrm{m}$
12:   **end for**
13:   $\boldsymbol{\theta} \leftarrow \mathrm{SGD}(\mathcal{D}, f, \boldsymbol{\theta}, \mathbf{s}, \mathbf{m}, \lceil \frac{S}{2} \rceil, \frac{\alpha}{10}, \mathcal{R})$
14:   **return** $\boldsymbol{\theta}, \mathbf{s}, \mathbf{m}$
15: **end function**

---

# B    ADDITIONAL RESULTS

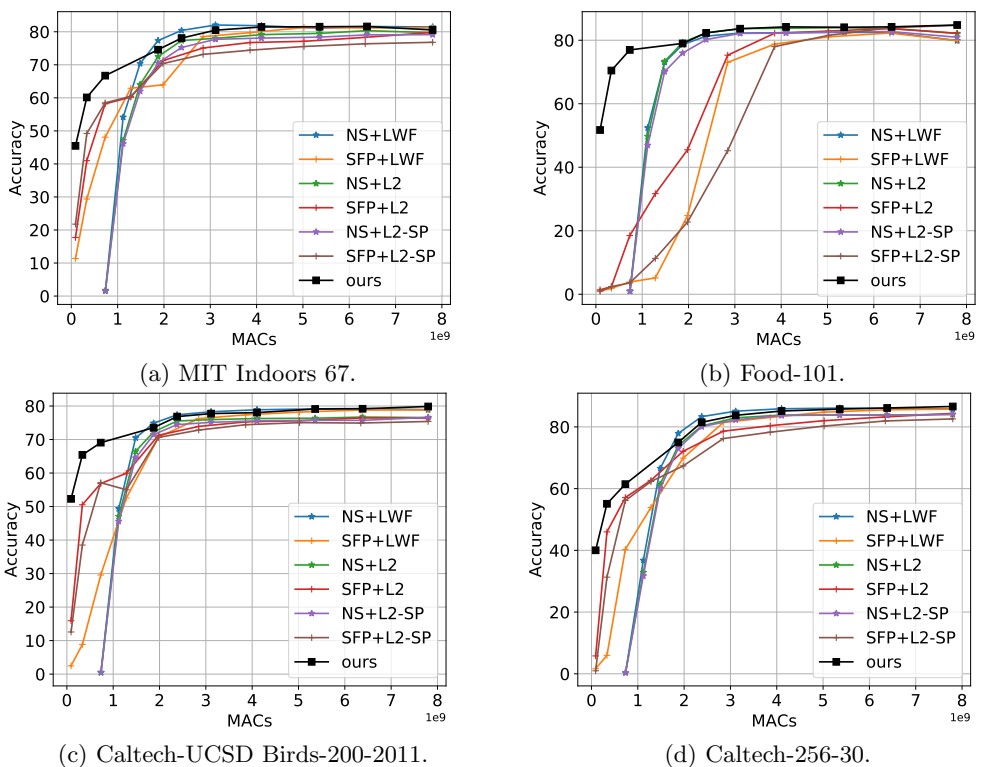

(a) MIT Indoors 67.

(b) Food-101.

(c) Caltech-UCSD Birds-200-2011.

(d) Caltech-256-30.

Figure 4: MACs and accuracy (%) trade-off comparisons among different joint methods.

