# OpenReview forum: "Pay Attention to Features, Transfer Learn Faster CNNs"
_ICLR.cc/2020/Conference — Accept (Poster)_

### Official Review · AnonReviewer3 · 2019-10-28
**Official Blind Review #3**

**Rating:** 6

**Review:**

 This paper proposes a method called attentive feature distillation and selection (AFDS) to improve the performance of transfer learning for CNNs. The authors argue that the regularization should constrain the proximity of feature maps, instead of pre-trained model weights. Specifically, the authors proposes two modifications of loss functions: 1) Attentive feature distillation (AFD), which modifies the regularization term to learn different weights for each channel and 2) Attentive feature selection (AFS), which modifies the ConvBN layers by predicts unimportant channels and suppress them.

Overall, this is a good work in terms of theory and experimentation, thus I would recommend to accept it. The approach is well motivated, and the literature is complete and relevant. The author's argument is validated by experiments comparing the proposed AFDS method and other existing transfer learning methods.

To improve this paper, the authors are suggested to address the following issues:
1.  Section 3.5 is not well organized. Besides, it is not mentioned what value the threshold hyper-parameter delta_m is set.
2. In page 9, "MACs" is missing in the sentence "In Figure 3, ...the number of vs. the target..."


**Experience Assessment:**

I have published one or two papers in this area.

**Review Assessment: Checking Correctness Of Derivations And Theory:**

I assessed the sensibility of the derivations and theory.

**Review Assessment: Checking Correctness Of Experiments:**

I assessed the sensibility of the experiments.

**Review Assessment: Thoroughness In Paper Reading:**

I read the paper at least twice and used my best judgement in assessing the paper.

---

> ### Author Response · Authors · 2019-11-11
> **Thank you for your detailed reviews.**
>
> Thank you for your comments. We would like to respond to the issues kindly raised by the reviewer:
> 1. In the last sentence of Section 3.5, we mentioned that ‘delta_s’ is set to a value such that 50% of the channel neurons use the predictor function ‘h_l’”. For this we mean that we first compute the variances of  “h_l(x_{l-1})” for each channel, and use the median of the channel variances as the value of the threshold “delta_s”. As kindly suggested by the reviewer, we will be updating this section accordingly.
> 2. Thanks for pointing out this to us, it will be fixed in the next revision.

---

### Official Review · AnonReviewer2 · 2019-10-29
**Official Blind Review #2**

**Rating:** 6

**Review:**

In general, I think it is a good paper and I like the contribution of the author. I think they explain in detail the methodology. The results compare the new methodologies with different databases which increase the credibility of the results. However, there is a couple of additional question that is important to manage:

1) The paper presents three different contributions. However, it is so clear how this work helps for "By changing the fraction of channel neurons to skip for each convolution, AFDS can further accelerate the transfer learned models while minimizing the impact on task accuracy" I think a better explanation of this part it would be necessary.

2) The comparison of the results are very focused on AFDS, Did you compare the results with different transfer learning approach?

3) During the training procedure. We need a better explanation of why "we found that in residual networks with greater depths, AFS could become notably challenging to train to high accuracies". Also, the results of the empirical test it would be useful to understand the challenges to train the network.

4) I think it would be useful to have the code available for the final version.

**Experience Assessment:**

I have published one or two papers in this area.

**Review Assessment: Checking Correctness Of Derivations And Theory:**

I assessed the sensibility of the derivations and theory.

**Review Assessment: Checking Correctness Of Experiments:**

I assessed the sensibility of the experiments.

**Review Assessment: Thoroughness In Paper Reading:**

I read the paper at least twice and used my best judgement in assessing the paper.

---

> ### Author Response · Authors · 2019-11-11
> **Thank you for your detailed reviews.**
>
> Thank you for your comments. We would like to answer your questions:
> 1. Consider a convolution operation with a “k * k” kernel, which takes input features with “Ci” channels, and computes feature maps of shape “Co * Ho * Wo”. To evaluate the convolution thus requires “k^2 * Ci * Co * Ho * Wo” multiply-accumulate operations (MACs). AFS can reduce the number of MACs required in a coarse-grained manner: before computing the convolution, AFS can predict the importance of each output channel, request the convolution to evaluate only “ceil(d * Co)” channels, and skip the remaining channels by setting them to zeros. Note that if the preceding layer is also a convolution that produce sparse outputs with only “d * Ci” non-zero channels, the input channels can also be skipped, reducing the number of MACs required to “k^2 * d^2 * Ci * Co * Ho * Wo”, a quadratic reduction in terms of “d”. We will update Section 3.4 to explain this in greater detail.
> 2. As previous work did not examine the opportunity of pruning and transfer learning jointly, In Table 2, we re-implemented L2, L2-SP [1] and LwF [2]. We then used the best they can achieve with any one of the pruning methods, and compared the results against AFDS under 2x, 5x or 10x speedup constraints. We have additionally compared to existing smaller transfer learned models from related works [3, 4] in Table 3.
> 3. We suspect the primary reason for the challenge is with the initial weights used in AFS. As the network depth gets larger, small changes in the variance used in initialization would result in highly sensitive changes in gradient magnitudes. Thanks for pointing out this to us and we will look into this in greater detail and update the paper accordingly.
> 4. The code and accompanying models will be made available soon.
>
> [1]: Xuhong Li, et al., Explicit Inductive Bias for Transfer Learning with Convolutional Networks, ICML 2018.
> [2]: Zhizhong Li, et al., Learning without Forgetting, IEEE Transactions on Pattern Analysis and Machine Intelligence, 2018.
> [3]: Sergey Zagoruyko, Nikos Komodakis, Paying More Attention to Attention: Improving the Performance of Convolutional Neural Networks via Attention Transfer, ICLR 2017.
> [4]: Yunhun Jang, et al., Learning What and Where to Transfer, ICML 2019.

---

### Official Review · AnonReviewer1 · 2019-10-31
**Official Blind Review #1**

**Rating:** 8

**Review:**

The paper presents an improvement to the task of transfer learning by being deliberate about which channels from the base model are most relevant to the new task at hand. It does this by apply attentive feature selection (AFS) to select channels or features that align well with the down stream task and attentive feature distillation (AFD) to pass on these features to the student network. In the process they do channel pruning there by decreasing the size of the network and enabling faster inference speeds. Their major argument is that plain transfer learning is redundant and wasteful and careful attention applied to selection of the features and channels to be transfered can lead to smaller faster models which in several cases presented in the paper provide superior performance.

Paper is clear and concise and experimentally sound showing a real contribution to the body of knowledge in transfer learning and pruning.

**Experience Assessment:**

I have read many papers in this area.

**Review Assessment: Checking Correctness Of Derivations And Theory:**

I assessed the sensibility of the derivations and theory.

**Review Assessment: Checking Correctness Of Experiments:**

I assessed the sensibility of the experiments.

**Review Assessment: Thoroughness In Paper Reading:**

I read the paper at least twice and used my best judgement in assessing the paper.

---

> ### Author Response · Authors · 2019-11-11
> **Thank you.**
>
> We would like to thank the reviewer for the positive comments.

---

### Public Comment · ~Mert_Kilickaya1 · 2019-10-02
**Relevant work of Adaptive Inference Graphs ~ AIG (Veit and Belongie, ECCV 2018) ?**

Hello,

Nice work that learns to select which features from a source model will help improve the fine-tuning performance on a target task! Indeed many times only few features contribute to the recognition in the target domain and others can even act as a confuser.

In structured sparsity section, you mention a few works that try to select which inference path to evaluate conditioned on the input. I thought (Veit and Belongie, ECCV 2018, http://openaccess.thecvf.com/content_ECCV_2018/papers/Andreas_Veit_Convolutional_Networks_with_ECCV_2018_paper.pdf) is also relevant for this list since they are learning to skip or process a layer (not channels) in Residual Networks without introducing architectural changes, and is tested on ImageNet. In this manner this can also be a coarser-grained alternative to "channel skipping" network proposed in this work.

My question is: Did you also try to do layer-wise skipping? It could be intuitive that some classes in the source domain may not even exist in the target domain (esp. for Birds-only and Dogs-only datasets) so than deciding to not process a layer on its entirety may yield even bigger drop in number of FLOPS, while maybe even increasing the accuracy?

---

### Public Comment · ~Gladis_Ne_Limes1 · 2023-05-22
**re**

Read more in https://mlsdev.com/services/ui-ux-design
 1. UX/UI Design Interviews & Research: Consult with stakeholders and end users to create a thorough understanding of user needs, design goals, and project objectives. 2. User Personas & Use Cases: Generate user personas and use cases to focus the design process and scope requirements for the project. 3. User Interface & Interaction Design: Create high-fidelity wireframes, interface designs, and prototypes to bring user interfaces to life and enable user testing. 4. Interface Testing & Usability: Perform A/B testing, user-testing, and iterative usability testing to ensure design solutions meet user needs. 5. Accessibility & Responsive Design: Ensure design solutions are accessible

---

### Decision · Program_Chairs · 2019-12-19

**Decision:**

Accept (Poster)

**Comment:**

This paper presents an attention-based approach to transfer faster CNNs, which tackles the problem of jointly transferring source knowledge and pruning target CNNs.

Reviewers are unanimously positive on the paper, in terms of a well-written paper with a reasonable approach that yields strong empirical performance under the resource constraint.

AC feels that the paper studies an important problem of making transfer learning faster for CNNs, however, the proposed model is a relatively straightforward combination of fine-tuning and filter-pruning, each having very extensive prior works. Also, AC has very critical comments for improving this paper:

- The Attentive Feature Distillation (AFD) module is very similar to DELTA (Li et al. ICLR 2019) and L2T (Jang et al. ICML 2019), significantly weakening the novelty. The empirical evaluation should consider DELTA as baselines, e.g. AFS+DELTA.

I accept this paper, assuming that all comments will be well addressed in the revision.